# Subspace based Federated Unlearning

**Guanghao Li**                                                                    *ligh24@mails.tsinghua.edu.cn*
*Tsinghua Shenzhen International Graduate School, Tsinghua University*
*College of Engineering, Southern University of Science and Technology*

**Li Shen** *                                                                         *mathshenli@gmail.com*
*School of Cyber Science and Technology, Shenzhen Campus of Sun Yat-sen University*

**Yan Sun**                                                                         *ysun9899@uni.sydney.edu.au*
*School of Computer Science, Faculty of Engineering, The University of Sydney*

**Yue Hu**                                                                               *huyue11@nudt.edu.cn*
*College of Systems Engineering, National University of Defense Technology*
*State Key Laboratory of Digital Intelligent Modeling and Simulation*

**Han Hu**                                                                                     *hhu@bit.edu.cn*
*School of Information and Electronics, Beijing Institute of Technology*

**Dacheng Tao**                                                                       *dacheng.tao@gmail.com*
*College of Computing and Data Science, Nanyang Technological University*

**Reviewed on OpenReview:** *https://openreview.net/forum?id=KE2ZNl2lFP*

## Abstract

Federated learning (FL) enables collaborative machine learning among multiple clients while preserving user data privacy by preventing the exchange of local data. However, when users request to leave the FL system, the trained FL model may still retain information about their contributions. To comply with the right to be forgotten, federated unlearning has been proposed, which aims to remove a designated client's influence from the FL model. Existing federated unlearning methods typically rely on storing historical parameter updates, which may be impractical in resource-constrained FL settings. In this paper, we propose a Subspace-based Federated Unlearning method (SFU) that addresses this challenge without requiring additional storage. SFU updates the model via gradient ascent constrained within a subspace, specifically the orthogonal complement of the gradient descent directions derived from the remaining clients. By projecting the ascending gradient of the target client onto this subspace, SFU can mitigate the contribution of the target client while maintaining model performance on the remaining clients. SFU is communication-efficient, requiring only one round of local training per client to transmit gradient information to the server for model updates. Extensive empirical evaluations on multiple datasets demonstrate that SFU achieves competitive unlearning performance while preserving model utility. Compared to representative baseline methods, SFU consistently shows promising results under various experimental settings.

## 1 Introduction

The traditional training approach of deep learning typically aggregates data from various participants. However, certain data, such as medical records [31], cannot be relocated from the hospital due to concerns regarding data privacy and individual preferences. In response, Federated Learning (FL) [32; 25; 3; 28]

---

*Corresponding author

emerges as a prominent decentralized machine learning solution for addressing these challenges. FL facilitates the training of a global model by exchanging model parameters between clients and a central server, effectively bypassing the need to transfer the raw data [23; 24; 28].

Recent privacy legislations [5; 35; 40] provide data owners the right to be forgotten. In the context of machine learning, this right necessitates two key actions: (i) deletion of user data from the storing entity and (ii) removal of the data's influence on the model [15]. Within the realm of Federated Learning (FL), federated unlearning crystallizes the embodiment of the right to be forgotten. However, achieving machine unlearning within the framework of federated learning presents heightened challenges: (1) **Limited Data Access**: In FL, the server lacks direct access to all data and associated operations, rendering forgetfulness techniques reliant on dataset segmentation inapplicable to FL scenarios. (2) **Multi-client Participation**: The initial model of each client in every training round depends on aggregating models from clients engaged in prior-round training, resulting in the gradual propagation of effects from individual data samples across models used for local training at multiple clients [34; 33; 41]. Thus, erasing data samples from one client requires a substantial number of clients to engage in a retraining process.

As mentioned above, retraining in FL demands a significant number of clients to participate in local training, inevitably leading to extended training durations. Some recent endeavors have been focused on addressing this challenge, such as storing the client's historical updated gradient data on the server and utilizing it to revert the trained global model [46; 30]. However, these methods necessitate either the client or the server to retain additional data or gradient information, which may not be practical in FL scenarios with limited storage resources. There are alternative approaches to execute the unlearning process by directly modifying the final model. For instance, directly employing gradient ascent on the target client can achieve the immediate reduction of client data influence in the final model. Nevertheless, this approach may considerably compromise the model's performance.

In this paper, we focus on developing a practical approach for implementing federated unlearning within the final model. We consider unlearning as the inverse process of learning via gradient ascent. However, the loss function is unbounded, requiring constraints on the gradient to preserve model quality [8; 6]. Consequently, we treat the entire process as a constraint-solving problem, aiming to maximize the empirical loss of the target client while maintaining acceptable model performance for other clients. Introducing updates orthogonal to the gradient directions of neural network predictions can induce minimal changes in network output [10]. Building on this idea, we propose a Subspace-based Federated Unlearning method, termed SFU.

In the SFU framework, the server only requires the ascending gradient information from the target client and the descending gradient information from the remaining clients. The server then projects the ascending gradient onto the orthogonal subspace of the descending gradient space. This constrained gradient is used to update the final trained FL global model, aiming to reduce the contribution of the target client while preserving the model's utility. Fig. 1 illustrates the core idea of our approach.

Specifically, participants in the SFU process can be categorized into three roles: the target client to be forgotten, the remaining clients, and the server. The target client performs local gradient ascent based on the global model and sends the gradient to the server. Each remaining client performs gradient descent and transmits their respective gradients to the server. The server aggregates the descending gradients and derives the gradient space through Singular Value Decomposition (SVD) [16; 37]. Finally, the server projects the ascending gradient onto the orthogonal complement of the others' gradient subspace and updates the global model. The SFU framework only requires one round of local training per client and a subsequent server update, without the need to access each client's raw data or store historical gradients. Empirical results indicate that SFU achieves competitive unlearning performance across various datasets while maintaining reasonable model accuracy.

In conclusion, our main contributions are as follows:

- We incorporate subspace learning into federated unlearning and propose a novel algorithm named SFU, which performs gradient ascent within a subspace that is orthogonal to the gradient space of the remaining clients, thereby explicitly constraining the unlearning update to reduce collateral damage.

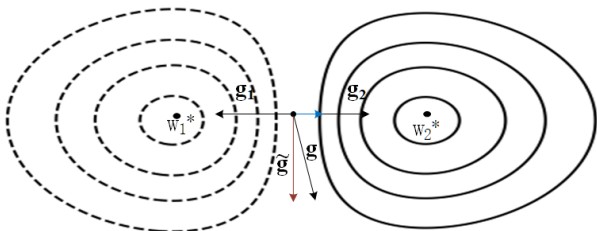

Figure 1: An illustration of SFU with three clients. $w_1^*$ and $w_2^*$ represent the optimal model parameters for client 1 and client 2, respectively. The ascending gradient of the target client is denoted as $g$. Additionally, $g_1$ and $g_2$ correspond to the descending gradients originating from client 1 and client 2. The projection of $g$ onto the orthogonal space spanned by $g_1$ and $g_2$ is denoted as $\tilde{g}$. Operating within this orthogonal subspace ensures minimal perturbation to the model's performance on client 1 and client 2.

- SFU achieves effective unlearning while maintaining acceptable utility on the remaining clients, and it does so without requiring additional storage of historical client updates or intermediate checkpoints, leading to no extra storage overhead.

- We conduct comprehensive experiments to evaluate SFU under different training stages and system conditions (e.g., varying client participation rates and target-client data volumes), demonstrating that SFU achieves competitive performance across multiple datasets, including MNIST, CIFAR10, and CIFAR100.

## 2 Related Work

**Machine unlearning.** The concept of "machine unlearning" entails the complete removal of a specific training data instance, necessitating the nullification of its impact on both extracted features and models. The introduction of machine unlearning is attributed to Cao & Yang [7], who reformulate statistical query learning into a summation form and achieve unlearning by selectively updating a subset of the summation. However, this algorithm is confined to transformable traditional machine-learning methods, prompting exploration into machine unlearning for various ML models. Ginart et al. [12] formalize the notion of effective data deletion in machine learning and propose two efficient deletion strategies for the K-means clustering algorithm. In the realm of supervised linear regression, Izzo et al. [17] develop the projective residual update (PRU) for linear and logistic models. While the computational cost of PRU scales linearly with feature dimensions, its suitability is limited for more intricate models such as neural networks. To address the overhead of forgetting, Bourtoule et al. [4] introduce the versatile SISA framework, which trains disjoint sub-models on data shards and only retrains the affected shard after a deletion request. A recent survey [18] provides a comprehensive overview of progress, but most studies still assume centrally accessible data, which is ill-suited for federated settings.

**Federated unlearning.** Current work can be grouped into two main directions. *Retraining-efficient methods.* Liu et al. [30] cache historical gradients to accelerate retraining; Su & Li [42] cluster clients and retrain only within the group containing the target client, which is useful in highly heterogeneous scenarios. *One-shot model modification.* Wu et al. [46] subtract stored gradients of the target client from the final model, still requiring extra storage. Wang et al. [45] prune task-specific weights, but only support class-level forgetting. Very recently, NoT [20] introduces a storage-free weight-negation strategy that supports client-, class-, and instance-wise unlearning without retraining. The definition of exact federated unlearning was formalized in FATS [44], which provides a TV-stable FedAvg variant and a provable closed-form unlearning step equivalent to retraining—but it requires server-side batch data and complex aggregation schemes.

Our approach is closely related to UPGA [15] and EWC-SAG [47]. All three perform gradient ascent on the final model to remove a client's influence, but differ in their constraints: UPGA constrains the *magnitude* of the update with an $\ell_2$ ball; EWC-SAG employs an elastic weight penalty; in contrast, **SFU** constrains the *direction* of the update to the orthogonal complement of the remaining clients' descent subspace, eliminating the need for historical checkpoints while better preserving accuracy.

## 3    Method

We propose *Subspace-based Federated Unlearning* (SFU), summarized in Algorithm 1. SFU leverages constrained gradient information from the target client to adapt the final trained model, effectively removing client contributions while upholding model performance across other clients. Notably, this method eliminates the need for the server to retain a historical record of parameter updates from individual clients and obviates the necessity for extensive retraining.

### 3.1    Problem Setup

Suppose there are $N$ clients, denoted as $C_1, \ldots, C_N$. Let $[N] := \{1, \ldots, N\}$. Client $C_i$ possesses a local dataset $\mathcal{D}^i$. The objective of conventional Federated Learning (FL) is to collaboratively train a machine learning model $w$ over the combined dataset $\mathcal{D} \triangleq \bigcup_{i \in [N]} \mathcal{D}^i$:

$$w^* \triangleq \arg \min_w \ L(w) \ = \ \sum_{i=1}^{N} \frac{|\mathcal{D}^i|}{|\mathcal{D}|} L_i(w), \tag{1}$$

where $L_i(w) = \mathbb{E}_{(x,y) \sim \mathcal{D}^i}[\ell_i(w; (x,y))]$ represents the empirical loss of client $C_i$. Throughout the federated training process, each client minimizes their respective empirical risk $L_i(w)$. The model $w^*$ is the final outcome of the FL procedure.

Now let's delve into the strategy for eliminating the contribution of the target client $C_I$. An intuitive approach is to escalate the empirical risk $L_I(w)$ associated with the target client $C_I$, essentially reversing the learning process. However, a straightforward maximization of the loss will impact the model's performance on other clients. Federated unlearning needs to forget the contribution of the target client $C_I$ while ensuring the overall model performance. Consequently, federated unlearning can be formulated as follows:

$$\arg \max_w L_I(w) = \mathbb{E}_{(x,y) \sim \mathcal{D}^I}[\ell_I(w; (x,y))]$$
$$\text{s.t.} \qquad \mathcal{E}(w^*) - \mathcal{E}(w) \leq \delta \tag{2}$$

Here, $\delta$ represents a tolerable difference in model performance, while $\mathcal{E}(w)$ signifies the accuracy of model $w$ evaluated on the remaining clients within the FL system. Prior work by Halimi et al. [15] utilized the parameter distance between $w$ and $w^*$ as a constraint, although this parameter distance might not fully capture the performance disparity among different models. Conversely, the constraint presented in Eq. 2 effectively addresses this concern.

### 3.2    Subspace-based Federated Unlearning (SFU)

When applying the ascending gradient update of the target client to the global model without considering other clients, there is a high probability that the neural network will turn into a stochastic model. SFU restricts model updates to orthogonal subspaces aligned with the gradients of other clients to mitigate this issue. This approach achieves the goal of forgetting the target client's contribution while minimizing potential disruptions to model performance for other clients. The training process of SFU is depicted in Fig. 2. Participants in SFU encompass the target client, remaining clients, and the server. The target client employs gradient ascent and transmits the resulting gradient to the server. Other clients engage in gradient descent and forward their descending gradients to the server. The server calculates the gradient space of the other clients and performs unlearning updates on the global model. Subsequently, we will provide a more detailed explanation of the SFU process.

#### 3.2.1    Local training on clients

Satisfying the constraints outlined in Eq.(2) necessitates imposing restrictions on the ascending gradient. Drawing inspiration from Fig. 1, it becomes evident that moving orthogonally to the gradient space yields the least impact (or even negligible change locally) on the FL model's performance for clients. This valuable

---

**Algorithm 1** Subspace-based Federated Unlearning (SFU)

---

**Input:** FL global model $w^*$, local dataset $\mathcal{D}^i$ of client $i$, learning rate $\eta$.

1: **Target client $C_I$:**
2: $g_I \leftarrow \nabla L_I^{ul}(w^*)$
3: Send $g_I$ to the server
4: **Other clients:**
5: **for** each client $i \neq I$ **do**
6: $\quad g_i \leftarrow \nabla L_i(w^*)$
7: $\quad$ Send $g_i$ to the server
8: **Server:** Let $\mathcal{I}_{\text{others}} \subseteq [N] \setminus \{I\}$ denote participating others, $S := |\mathcal{I}_{\text{others}}|$.
9: **for** each network layer $l = 1, 2, \ldots, L$ **do**
10: $\quad R^l \leftarrow [\, \text{vec}(g_i^l)\,]_{i \in \mathcal{I}_{\text{others}}} \in \mathbb{R}^{r_l \times S}$
11: $\quad$ Perform SVD on $R^l = U^l \Sigma^l (V^l)^\top$
12: $\quad$ Choose the smallest $k$ s.t. $\|R_k^l\|_F^2 \geq \epsilon^l \|R^l\|_F^2$ and let $U_k^l = [u_1^l, \ldots, u_k^l]$
13: $\quad$ Projection matrix $P^l \leftarrow U_k^l (U_k^l)^\top$
14: $P \leftarrow \text{diag}(P^1, \ldots, P^L)$
15: $g_P \leftarrow (I - P)\, g_I$
16: $w_{ul} \leftarrow w^* - \eta g_P$

---

insight prompts us to project the updated gradient of the target client $C_I$ onto the orthogonal space of the gradient subspace associated with other clients [10]. This process requires both the ascending gradient from the target client and the descending gradients from the other clients. Hence, upon receiving a forget request from the target client, the training procedures for the target client and the other clients are as follows:

- **Target Client:** The target client performs gradient *ascent* on its local loss. To implement ascent with standard descent machinery, we minimize the surrogate $L_I^{ul}(w) := 1/L_I(w)$. Concretely, during unlearning we evaluate at the final FL model $w^*$ and compute a mini-batch stochastic gradient $g_I := \nabla L_I^{ul}(w^*)$, so $g_I$ is a negative multiple of $\nabla L_I(w^*)$ and descending along $g_I$ increases $L_I$. Hence the local objective for the target client is to minimize $L_I^{ul}(w)$. The target client then transmits the gradient information $g_I$ to the server.

- **Other Clients:** The remaining available clients follow the same gradient *descent* procedure as in standard FL; that is, each minimizes its local objective $L_i(w)$. During the unlearning round, for each $i \neq I$ we evaluate at the final FL model $w^*$ a mini-batch stochastic gradient $g_i := \nabla L_i(w^*)$, computed on client $i$'s data. Each such client then transmits $g_i$ to the server.

### 3.2.2 Computation of projection matrix

When the server aggregates the gradient information sent by each client, it proceeds to compute the projection matrix of descending gradient space for each layer of the model. This process involves two key steps:

- Let $L$ denote the total number of network layers. For each layer $l$, the per-client gradient can be viewed as a matrix $g_i^l \in \mathbb{R}^{m_l \times n_l}$. Let $\text{vec}(g_i^l) \in \mathbb{R}^{r_l}$ denote its vectorization (with $r_l = m_l n_l$). To form the gradient subspace at layer $l$, the server column-stacks the vectorized gradients from the available non-target clients $\mathcal{I}_{\text{others}} \subseteq [N] \setminus \{I\}$, yielding $R^l := [\, \text{vec}(g_i^l)\,]_{i \in \mathcal{I}_{\text{others}}} \in \mathbb{R}^{r_l \times S}$, where $S := |\mathcal{I}_{\text{others}}|$.

- Next, compute the SVD of $R^l$, i.e., $R^l = U^l \Sigma^l (V^l)^\top$. Based on a user-specified coverage coefficient $\epsilon^l \in (0, 1]$, choose the smallest $k$ and form the rank-$k$ approximation $R_k^l := U_k^l \Sigma_k^l (V_k^l)^\top$, where $U_k^l = [u_1^l, \ldots, u_k^l]$, $\Sigma_k^l = \text{diag}(\sigma_1^l, \ldots, \sigma_k^l)$, and $V_k^l = [v_1^l, \ldots, v_k^l]$, such that

$$\|R_k^l\|_F^2 \geq \epsilon^l \|R^l\|_F^2. \tag{3}$$

We then take the first $k$ left singular vectors to represent the significant subspace at layer $l$, i.e., $\text{span}\{u_1^l, \ldots, u_k^l\}$, with $U_k^l$ as its basis matrix, which captures the directions associated with the largest singular values [38].

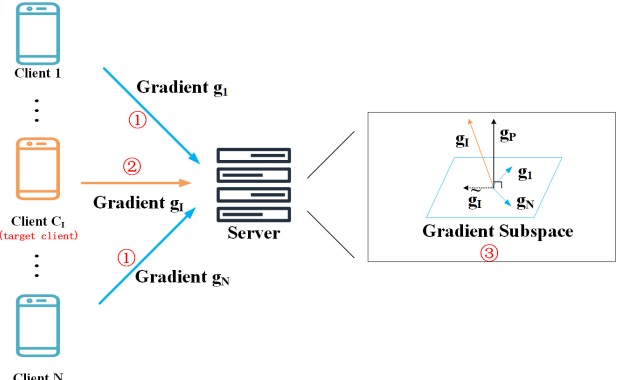

Figure 2: The pipeline of SFU. The entire process occurs subsequent to the training of the FL model. The orange client denotes the target client from which the contribution is to be removed; the blue clients represent others. The boxes on the right of the image symbolize global model updates performed on the server.

Following these steps, the server forms the orthogonal projector onto the others' gradient subspace at layer $l$ as $P^l := U_k^l (U_k^l)^\top$, with $(U_k^l)^\top U_k^l = I_k$. Repeating this for all layers yields a block-diagonal projector over the whole network, $P := \mathrm{diag}(P^1, \ldots, P^L)$.

### 3.2.3 Update of the global model on the server

After receiving the client gradients and constructing $P$, the server updates the global model $w^*$. First project the target client's gradient onto the others' gradient subspace, $\tilde{g}_I := P\, g_I$, and let the orthogonal component be $g_P := (I - P)\, g_I$. The server then updates in the orthogonal complement:

$$w_{ul} \;=\; w^* - \eta\, g_P, \tag{4}$$

where $\eta > 0$ is the learning rate. This update removes the target client $C_I$'s contribution while limiting interference with directions favored by the remaining clients, thereby preserving overall performance.

**Remarks.** Subspace methods have been broadly applied in continual learning [38], meta learning [19], one-shot learning [43], adversarial training [27], graph forgetting [9], and speeding up deep model training [26]. To the best of our knowledge, **SFU** is the first to exploit updates in the orthogonal complement of the remaining clients' gradient subspace for federated unlearning.

### 3.3 On the effectiveness of SFU

The effectiveness of SFU is twofold: it preserves overall performance on the remaining clients while enforcing forgetting for the target client. In this section, we analyze its efficacy. Let $w_r$ denote the model obtained by retraining *from scratch*, *without* the target client, starting from the same initialization $w(0)$. Let $T$ be the total number of gradient updates (equivalently, the aggregated local epochs) used in FL training. Let $w_{ul}$ denote the model returned by a single application of SFU to the final FL model $w^*$.

**Performance preservation.** We aim to show that $\|w_r - w_{ul}\|_2$ is bounded, thereby ruling out pathological deviations. Before presenting the result, we state the following standard assumptions on stochastic gradients, collected in Assumption 1; these are commonly used in the theoretical analysis of federated learning [50; 29].

**Assumption 1.** *The expected squared norm of stochastic gradients is uniformly bounded for all $w$:* $\mathbb{E}\|\nabla L(w)\|_2^2 \leq G^2$ *(hence, $\mathbb{E}\|\nabla L(w)\|_2 \leq G$). The same bound holds for retraining gradients $\nabla L^r(w)$, where $L^r(w) := \sum_{i \in [N] \setminus \{I\}} \frac{|\mathcal{D}^i|}{|\mathcal{D}|} L_i(w)$. Moreover, the target-client gradient is bounded: $\mathbb{E}\|g_I\|_2 \leq G$.*

**Theorem 1.** *Under Assumption 1, let $w_{ul} = w^* - \eta\, g_P$ be the model returned by a single SFU update, and let $w_r$ be the model obtained by retraining from scratch (without the target client) from the same initialization after $T$ updates. Then*

$$\mathbb{E}\, \|w_r - w_{ul}\|_2 \;\leq\; (2\eta T + \eta)\, G \,. \tag{5}$$

This shows that the deviation between the SFU-updated model and the retrained model is bounded.

*Proof.* Let us define $w(t)$, $w_r(t)$ as the weight parameters obtained by training from scratch with target client and without target client for $t$ epochs with the same initialization $w(0) = w_r(0)$, and $w_{ul}$ denote the weight parameters obtained by applying SFU on $w^*$. To help the reader better understand the proof strategy used in this section, we derive the upper bound of $\mathbb{E}\|w_r(T) - w_{ul}\|_2$ by first expanding the formula into two terms

$$\mathbb{E}\|w_r(T) - w_{ul}\|_2 \leq \mathbb{E}\|w_r(T) - w(T)\|_2 + \mathbb{E}\|w(T) - w_{ul}\|_2. \tag{6}$$

**First term.** By the update rules $w_r(t+1) = w_r(t) - \eta \nabla L^r(w_r(t))$ and $w(t+1) = w(t) - \eta \nabla L(w(t))$, where $L^r(w) = \sum_{i \in [N] \setminus \{I\}} \frac{|\mathcal{D}^i|}{|\mathcal{D}|} L_i(w)$, we have

$$\begin{aligned} \mathbb{E}\|w_r(t+1) - w(t+1)\|_2 &\leq \mathbb{E}\|w_r(t) - w(t)\|_2 + \eta \mathbb{E}\|\nabla L^r(w_r(t))\|_2 + \eta \mathbb{E}\|\nabla L(w(t))\|_2 \\ &\leq \mathbb{E}\|w_r(t) - w(t)\|_2 + 2\eta G. \end{aligned} \tag{7}$$

Then after $T$ iterations, we can bound the difference between two parameters as

$$\mathbb{E}\|w_r(T) - w(T)\|_2 \leq 2\eta TG. \tag{8}$$

**Second term.** Since $w_{\text{ul}} = w(T) - \eta g_P$ with $g_P := (I - P)g_I$, it follows that $\|g_P\|_2 \leq \|g_I\|_2$. Therefore,

$$\mathbb{E}\|w(T) - w_{ul}\|_2 = \eta \mathbb{E}\|g_P\|_2 \leq \eta \mathbb{E}\|g_I\|_2 \leq \eta G. \tag{9}$$

Adding the bound of the first term and the bound of the second term we can obtain the final bound $(2\eta T + \eta)G$. $\qquad\square$

**Forgetting of the target client.** We now show that SFU enforces forgetting at the target: the update direction $-g_P$ is a descent direction for the surrogate loss $L_I^{ul}$ (equivalently, $g_P$ is an ascent direction for $L_I$).

**Theorem 2.** *Let $g_I := \nabla L_I^{ul}(w^*)$ and let $P$ be the orthogonal projector onto the gradient subspace spanned by the other clients. Define $\tilde{g}_I := P g_I$ and $g_P := (I - P) g_I$. Then $-g_P$ is a descent direction for $L_I^{ul}$ at $w^*$; in particular, $\langle -g_P, g_I \rangle = -\|g_P\|_2^2 \leq 0$.*

*Proof.* For a differentiable function $f$, a vector $u$ is a descent direction at $w$ if $\langle u, \nabla f(w) \rangle \leq 0$. Taking $f = L_I^{ul}$ and using that $P$ is an orthogonal projector, we have $g_P \perp \tilde{g}_I$; hence

$$\begin{aligned} \langle -g_P, g_I \rangle &= \langle -g_P, g_P + \tilde{g}_I \rangle \\ &= -\|g_P\|_2^2 - \langle g_P, \tilde{g}_I \rangle = -\|g_P\|_2^2 \leq 0. \end{aligned} \tag{10}$$

Therefore, $-g_P$ is a descent direction for $L_I^{ul}$ at $w^*$. $\qquad\square$

Together, the two results show that SFU updates the model in a direction that removes (forgets) the target client's contribution while respecting the subspace favored by the remaining clients. They also imply a bounded deviation between the SFU-updated model and the model obtained by full retraining without the target client, thereby guaranteeing that overall performance is preserved within a controlled margin.

## 4 Experiments

In this section, we empirically assess the effectiveness of SFU across various model architectures on three datasets. Our experimental findings demonstrate that our unlearning strategies can adeptly eliminate the target client's impact on the global model while incurring minimal performance degradation. Furthermore, these strategies facilitate rapid accuracy recovery within a few training rounds. We commence by outlining the experimental setup and subsequently unveil the evaluation outcomes. For comprehensive experimental settings, please consult the Appendix.

| Dataset | distribution | network | FedAvg | | Retraining | | UPGA | | ULS | | EWC-SGA | | SFU | |
|---|---|---|---|---|---|---|---|---|---|---|---|---|---|---|
| | | | test acc | atk acc | test acc | atk acc | test acc | atk acc | test acc | atk acc | test acc | atk acc | test acc | atk acc |
| MNIST | IID | MLP | 96.57 | 99.14 | 96.45 | 0.61 | 86.48 | 0.10 | 83.42 | 0.0 | 90.19 | 0.24 | **92.8** | 0.15 |
| | | CNN | 99.15 | 99.63 | 99.21 | 0.20 | 51.59 | 0.0 | 34.95 | 0.0 | 88.9 | 1.56 | **98.62** | 0.06 |
| | | ResNet | 99.56 | 98.32 | 99.51 | 0.51 | 51.80 | 0.0 | 45.09 | 0.0 | 89.27 | 1.24 | **99.03** | 0.06 |
| | Dir(0.6) | MLP | 96.03 | 99.04 | 95.95 | 0.60 | 81.91 | 0.14 | 68.41 | 0.0 | 86.69 | 1.41 | **89.12** | 0.25 |
| | | CNN | 98.96 | 99.58 | 99.01 | 0.26 | 14.11 | 0.0 | 46.43 | 0.0 | 85.25 | 0.0 | **98.76** | 0.01 |
| | | ResNet | 99.59 | 98.23 | 99.55 | 0.79 | 22.20 | 0.0 | 51.42 | 0.0 | 85.79 | 0.0 | **99.39** | 0.03 |
| | Dir(0.3) | MLP | 95.54 | 98.95 | 95.48 | 0.66 | 83.21 | 0.24 | 72.96 | 0.0 | 86.34 | 1.88 | **88.63** | 0.26 |
| | | CNN | 98.92 | 99.78 | 98.98 | 0.28 | 52.41 | 0.0 | 38.76 | 0.0 | 86.83 | 0.0 | **98.39** | 0.03 |
| | | ResNet | 99.41 | 98.39 | 99.38 | 0.69 | 55.67 | 0.10 | 42.95 | 0.0 | 85.26 | 1.24 | **98.88** | 0.03 |
| CIFAR10 | IID | MLP | 50.86 | 52.38 | 50.6 | 3.77 | 11.05 | 0.01 | 13.75 | 0.0 | 20.45 | 0.08 | **44.76** | 0.96 |
| | | CNN | 66.40 | 37.14 | 67.56 | 4.42 | 60.94 | 9.40 | 10.0 | 0.0 | 61.48 | 1.02 | **61.68** | 0.46 |
| | | ResNet | 83.73 | 38.27 | 83.98 | 7.23 | 76.84 | 8.56 | 12.61 | 0.0 | 77.53 | 0.84 | **78.06** | 0.46 |
| | Dir(0.6) | MLP | 47.58 | 53.58 | 48.42 | 4.81 | 16.54 | 0.02 | 11.97 | 0.0 | 33.4 | 0.12 | **42.95** | 3.67 |
| | | CNN | 63.74 | 39.88 | 65.83 | 3.21 | 64.2 | 8.34 | 53.27 | 0.85 | 56.27 | 0.94 | **61.52** | 6.43 |
| | | ResNet | 80.79 | 46.66 | 81.62 | 2.49 | 75.40 | 7.52 | 67.44 | 0.0 | 71.64 | 0.94 | **77.95** | 2.82 |
| | Dir(0.3) | MLP | 46.41 | 54.4 | 45.51 | 6.01 | 13.69 | 0.01 | 10.36 | 0.0 | 30.73 | 0.26 | **45.21** | 1.53 |
| | | CNN | 61.52 | 51.84 | 63.51 | 4.54 | 38.76 | 0.08 | 10.06 | 0.0 | 50.07 | 1.43 | **54.67** | 3.72 |
| | | ResNet | 79.91 | 56.87 | 79.55 | 1.59 | 50.35 | 0.08 | 13.07 | 0.0 | 65.04 | 2.31 | **71.01** | 1.56 |
| CIFAR100 | IID | MLP | 21.93 | 57.79 | 23.42 | 1.73 | 9.05 | 0.01 | 2.03 | 0.0 | 20.01 | 0.26 | **22.06** | 1.46 |
| | | CNN | 33.20 | 44.66 | 33.68 | 0.48 | 34.35 | 0.13 | 1.75 | 0.0 | 31.50 | 0.20 | **32.45** | 1.27 |
| | | ResNet | 58.02 | 50.38 | 57.75 | 0.81 | 60.03 | 0.11 | 3.06 | 0.0 | 54.05 | 0.18 | **55.71** | 1.36 |
| | Dir(0.6) | MLP | 21.12 | 50.89 | 21.8 | 0.78 | 9.72 | 0.0 | 1.59 | 0.0 | 19.12 | 0.50 | **20.12** | 0.71 |
| | | CNN | 31.26 | 47.77 | 32.06 | 0.58 | 29.38 | 0.02 | 28.52 | 0.03 | 30.89 | 0.09 | **31.55** | 0.12 |
| | | ResNet | 56.50 | 49.04 | 56.38 | 0.51 | 53.10 | 0.05 | 11.83 | 0.0 | 50.42 | 0.12 | **55.24** | 0.12 |
| | Dir(0.3) | MLP | 20.17 | 57.93 | 20.67 | 1.30 | 6.94 | 0.01 | 1.06 | 0.0 | 16.52 | 0.61 | **19.2** | 1.34 |
| | | CNN | 31.61 | 45.81 | 32.02 | 0.65 | 31.96 | 0.18 | 1.0 | 0.0 | 30.86 | 0.20 | **31.12** | 2.10 |
| | | ResNet | 55.11 | 50.87 | 55.70 | 0.50 | 45.72 | 0.26 | 1.74 | 0.0 | 51.08 | 0.82 | **54.56** | 1.92 |

Table 1: Results of different unlearning methods. We record the attack accuracy as "atk acc," and "test acc" represents the accuracy metric on the clean test data.

## 4.1 Experimental Setup

**Datasets.** We evaluate the performance of SFU using three widely recognized datasets: MNIST [22], CIFAR10, and CIFAR100 [21]. The datasets exhibit increasing levels of training difficulty from MNIST to CIFAR100. Our evaluation encompasses two distinct data distribution scenarios: Independent and Identically Distributed (IID), as well as Non-IID (Non Independent and Identically Distributed). For the Non-IID setting, we adopt the Dirichlet distribution ($\beta$): The label distribution on each device follows the Dirichlet distribution, where $\beta$ serves as the concentration parameter ($\beta > 0$).

**Baselines.** Our goal is to achieve the unlearning process by adapting the FL model. To this end, we select three prominent federated unlearning algorithms that *directly modify the final FL model* as reference benchmarks: Unlearning via Projected Gradient Ascent (UPGA) [15], UL-Subtraction (ULS) [46], and EWC-SGA[1] [47]. Furthermore, we include a full *retraining from scratch* baseline for comparative evaluation. These methods are chosen because they (i) require no multi-epoch retraining of all clients, (ii) do not assume access to auxiliary server-side data, and (iii) represent three different constraint strategies—$\ell_2$-ball, gradient subtraction, and elastic weight regularisation—thus providing a diverse yet fair yardstick for SFU.

**Unlearning Assessment Approaches.** We employ backdoor attacks during the target client's updates to the global model, enabling us to intuitively examine the unlearning impact through the success rate of a model backdoor attack on the unlearned global model. An effective unlearning method should diminish the success rate of such an attack post unlearning. It is worth noting that due to model prediction errors, even retraining can yield an attack success rate greater than 0. In our experiments, we follow the approach of Halimi et al. [15] and execute the backdoor attack using a "pixel pattern" trigger of size $2 \times 2$, altering the label to "9" on data with labels other than "9". Furthermore, we evaluate the model's performance after unlearning using accuracy metrics on untainted test data.

**Implementation details.** We evaluated three network architectures—MLP, CNN, and ResNet—on the MNIST, CIFAR-10, and CIFAR-100 datasets. These models were implemented using PyTorch [36]. Our experimental setup consists of 10 clients, with one designated as the target client. All clients fully participate in each training round. We backdoor 80% of the data on the target client. The hyperparameter settings for

---

[1]We follow the notation of Wu et al. [47] and denote *elastic weight consolidation with single ascent gradient* as EWC-SGA.

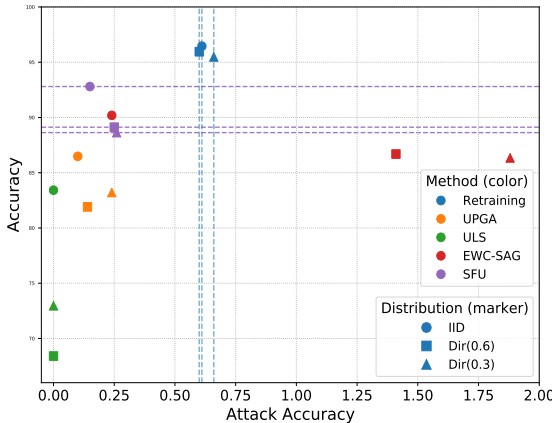

Figure 3: MNIST MLP: Accuracy vs Attack Accuracy.

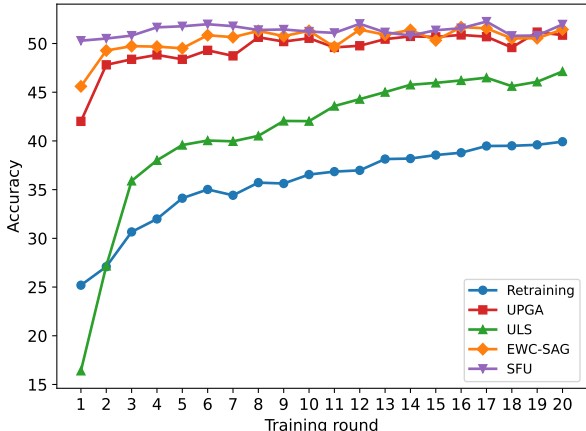

Figure 4: Convergence plots for SFU and other baselines on CIFAR-10 (CNN, IID).

each approach were as follows: In the case of SFU, the learning rate was set to 0.01, the number of epochs was set to 1, and a mini-batch size of 128 was utilized for gradient ascent on the target client. We explored the SVD parameter $\epsilon$ within the range $[0.90 - 0.99]$ and selected the optimal value. SFU constructs the gradient space by aggregating gradient information from all other clients. For UPGA and EWC-SGA, the learning rate, the number of epochs, and the mini-batch size were maintained consistent with those of SFU. Additionally, we performed a parameter search to determine the specific values of their unique parameters. All experiments were run on a single NVIDIA RTX 4090 (24 GB) GPU with 120 GB RAM.

## 4.2 Main Results

**Efficient unlearning with minimal performance loss.** Table 1 summarizes the unlearning performance of SFU and representative baselines across IID datasets and different model architectures. Comparable to full retraining, SFU consistently suppresses the backdoor attack success rate (ASR) to a low level on all evaluated benchmarks, indicating effective removal of the target client's malicious influence. Although several baselines can also reduce ASR, they frequently achieve this by substantially degrading the clean classification accuracy, reflecting an unfavorable utility–unlearning trade-off.For instance, on MNIST with the CNN backbone, UPGA reduces the attack performance but causes an accuracy drop of roughly 47 percentage points, whereas SFU maintains nearly the original utility with only an $\approx 1$ percentage-point decrease. Overall, these results demonstrate that SFU can achieve strong unlearning efficacy while preserving clean performance, delivering a more efficient and practical unlearning solution under minimal performance loss.

**Test–Attack accuracy trade-off.** Figure 3 plots test accuracy ($\uparrow$) against attack accuracy ($\downarrow$) for all methods on MNIST with an MLP. Owing to stochasticity, even full retraining can yield a non-zero attack accuracy. We therefore take the retraining point as a reference and deem a method to achieve successful forgetting if its attack accuracy is at or below the retraining attack accuracy (within a small tolerance $\delta$). Equivalently, points lying to the left of the retraining reference satisfy the forgetting requirement, and among those, higher test accuracy is preferable (i.e., the upper-left region). Under this criterion, SFU lies on or near the empirical Pareto frontier: it delivers the highest test accuracy while keeping attack accuracy within an acceptable range, indicating a superior balance between accuracy retention and attack mitigation compared with competing baselines.

**Robustness Across Different Data Settings.** The heterogeneity of data across different clients and the complexity of training tasks can both impact the federated learning process. Table 1 illustrates the influence of these factors on the unlearning process. It is evident that as the data heterogeneity increases, various unlearning methods lead to more significant performance degradation. For instance, during the transition from IID data to Dir(0.3) data, SFU's performance degradation on the CNN model trained on the CIFAR10 dataset increases from 4.72% to 6.85%. However, this performance change remains smaller than the performance losses observed in other algorithms when facing changes in data heterogeneity. Furthermore,

| Method | Target data | Extra comm. | Srv. storage | Public KD | Server comp. | Time |
|---|---|---|---|---|---|---|
| Retraining[†] | – | $T\,C_{\text{round}}$ | 0 | – | FL training | $T$ |
| UPGA | ✓ | $|w|$ | 0 | – | $O(|w|)$ | 1 |
| ULS | – | 0 | $\Theta(R|w|)$ | ✓ | $O(|w|) + \text{KD}(F)$ | $0+F$ |
| EWC-SGA | ✓ | $|w|$ | 0 | – | $O(|w|)$ (Fisher diag) | 1 |
| **SFU (ours)**[‡] | ✓ | $(S{+}1)|w|$ | **0** | – | per-layer SVD: $\tilde{O}\big(\sum_l r_l k_l^2\big)$ | **1** |

Table 2: **Resource comparison.** $C_{\text{round}}$: total uplink+downlink per FL round; $T$: # retraining rounds; $R$: # pre-unlearning rounds kept by ULS (history length); $F$: KD fine-tuning rounds; $|w|$: model size (one full model upload or download); $S$: # participating non-target clients. Here $r_l$ denotes the dimension of the vectorized gradient at layer $l$, $k_l$ the retained rank, and $\tilde{O}(\cdot)$ hides polylogarithmic factors. *Time* counts only the core unlearning round(s); optional KD for ULS is shown as $+F$. [†] All non-target clients participate *each round* (retraining). [‡] A *single collaborative round* with $S$ non-target clients (SFU).

as the task complexity shifts from MNIST to CIFAR100, some unlearning methods may become ineffective. For instance, ULS generates a random CNN model on the CIFAR10 dataset, while SFU manages to maintain the model's performance after unlearning. This highlights SFU's superior robustness across different data settings.

**Recovery of Model Accuracy After Unlearning.** After undergoing different Federated Unlearning methods, it is a common practice to further fine-tune the unlearned model on the remaining clients to restore its predictive utility. Accordingly, we fine-tuned the CNN models produced by each unlearning method on the remaining clients under the IID CIFAR-10 setting, using the same fine-tuning recipe to ensure a fair comparison. In contrast to these client-only recovery protocols, ULS additionally leverages a small amount of clean server-side data and performs knowledge-distillation-based fine-tuning, which provides extra supervision beyond the remaining clients. Fig. 4 summarizes the recovery curves on CIFAR-10 (IID). Notably, SFU exhibits a markedly faster and more effective recovery behavior: it reaches a higher accuracy after only one or a few retraining rounds, whereas other methods typically require substantially more rounds to approach similar performance. This indicates that SFU yields an unlearned model that remains closer to a high-utility solution on the retained data distribution, thereby serving as a stronger initialization for post-unlearning retraining and reducing the overall cost of accuracy recovery.

### 4.3 Resource implications and deployment trade-offs

Table 2 summarizes resource profiles using unified notation: $T$ (retraining rounds), $C_{\text{round}}$ (per-round uplink+downlink), $|w|$ (model size), $S$ (number of participating non-target clients), $R$ (length of stored history), and $F$ (KD fine-tuning rounds, if applicable). The *Time* column counts only the core unlearning round(s); optional post-processing (e.g., KD for ULS) is indicated as $+F$. Column abbreviations are: *Target data* (whether access to the target client's data is required), *Extra comm.* (additional communication beyond baseline FL, measured in multiples of $|w|$ or $C_{\text{round}}$), *Srv. storage* (extra server-side storage), *Public KD* (whether public-data knowledge distillation is required), and *Server comp.* (dominant server-side computation).

**Comparison summary.** Retraining serves as the time/communication baseline with $T$ rounds, no extra server storage, and per-round participation of all non-target clients; ULS incurs near-zero extra communication at unlearning time but requires storing $\Theta(R|w|)$ historical updates and typically relies on *public data* for knowledge distillation, adding $F$ rounds; UPGA/EWC-SGA require only the target client once (extra communication $\approx |w|$), need no other clients online, and impose $O(|w|)$-level server compute (EWC uses a Fisher-diagonal surrogate); SFU neither stores history nor relies on public data—its core cost is a single collaborative round in which the target uploads an ascent gradient and $S$ non-target clients upload descent gradients, followed by a per-layer (randomized) SVD at the server to project onto the orthogonal complement, with $(S{+}1)|w|$ extra communication and $\tilde{O}\big(\sum_l r_l k_l^2\big)$ compute.

**Practical takeaways.** Requiring target data at unlearning time is not inherently a disadvantage; it enables *client-controlled, fine-grained forgetting* (e.g., selecting specific subsets to forget) while keeping server-side

storage at zero. SFU is preferable when long-term storage of client histories is disallowed or undesirable, and when the target client can briefly be online with access to its data. In practice, the single-round cost of SFU can be further reduced by sampling a subset of non-target clients (reducing $S$) and by using layered/randomized SVD with an $\epsilon$-coverage criterion to choose per-layer ranks $k_l$.

### 4.4 Exploratory Study

**Impact of different SVD coefficient $\epsilon$.** As discussed in Section 3.2, our approach aims to mitigate the decline in clean accuracy by taking gradient steps orthogonal to the gradient subspace spanned by the remaining clients. This subspace is constructed from bases that approximate the dominant task representations of the remaining clients, and the extent of this approximation is controlled by the threshold hyperparameter $\epsilon^l$ at layer $l$, as defined in Eq. 3. Intuitively, $\epsilon$ determines how much gradient energy is retained when forming the subspace: a smaller $\epsilon$ keeps fewer principal directions (a lower-rank subspace), whereas a larger $\epsilon$ preserves more directions and better matches the remaining clients' gradient geometry. In our experiments, we maintain the same $\epsilon$ for all layers; Fig. 5 further demonstrates a consistent test–attack trade-off on CIFAR-10 (IID) across three architectures. Specifically, a lower $\epsilon$ value (closer to 0) yields a lower-rank retained subspace and leaves a larger orthogonal complement for the ascent update, which can remove the target client's influence more aggressively but may induce larger performance shifts on the remaining clients, resulting in reduced clean accuracy. Conversely, a higher $\epsilon$ value (closer to 1) preserves more gradient correlations and thus better maintains clean accuracy by restricting updates to directions less harmful to other clients, but it may not fully eliminate the target client's contribution, leading to increased attack success. Overall, Fig. 5 suggests that selecting $\epsilon$ in the range 0.9–0.95 strikes a good balance between effectively eliminating the target client's contribution and preserving the utility of the learned model.

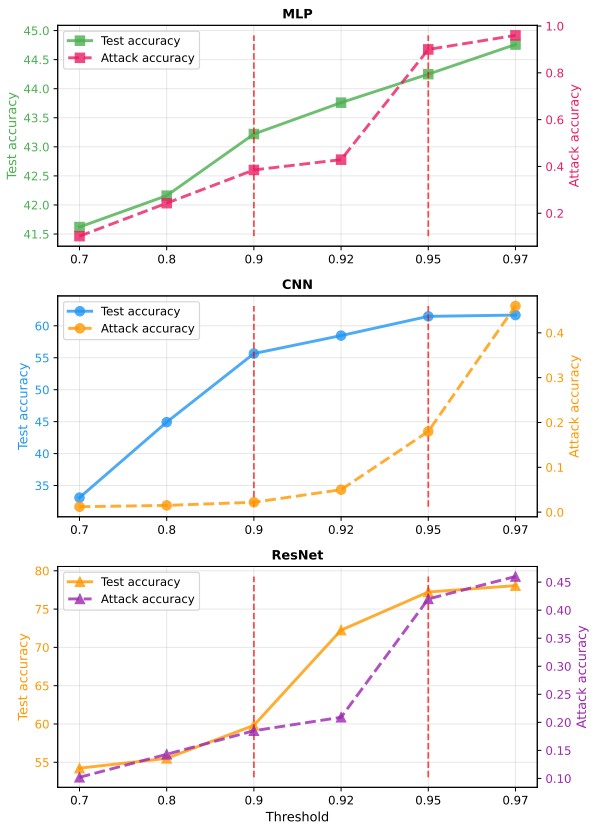

Figure 5: Effect of the SVD coverage coefficient $\epsilon$ on the test–attack trade-off (CIFAR-10, IID), evaluated with MLP/CNN/ResNet.

**Impact of client's participation rate.** In practical FL, not all clients are available in every round. We therefore vary the client participation rate, defined as the per-round availability (sampling) probability of clients, and evaluate how well SFU removes the target client while preserving the remaining clients' utility. As shown in Fig. 6(a), SFU consistently yields a favorable unlearning outcome across three architectures on CIFAR-10 (IID), and higher participation rates generally lead to better performance on the remaining clients, with the improvement gradually saturating as more clients become available. This trend is expected since a higher participation rate provides more non-target clients' descending gradients over rounds, resulting in a richer and more representative gradient subspace, which better constrains the unlearning update to preserve non-target directions while suppressing the target client's influence. In the extreme case of 0% participation, the system degenerates to purely local training without federated aggregation, and SFU cannot construct the orthogonal gradient space from non-target clients; consequently, applying the unlearning update without this subspace constraint leads to severe parameter drift and yields near chance-level accuracy (about 10% for 10-way classification). These results underscore the necessity of performing SFU within an orthogonal gradient space: even partial participation supplies sufficient geometric structure for meaningful unlearning, while higher participation further improves the utility of the resulting model.

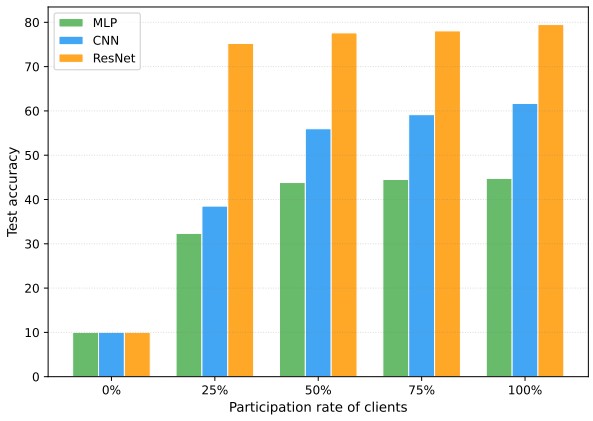 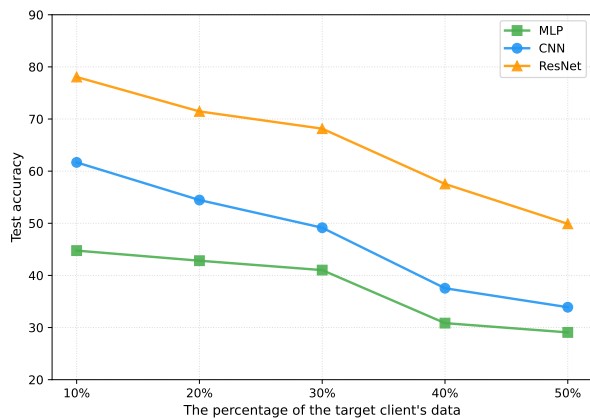

(a) Test accuracy (CIFAR-10, IID) vs. client participation rate.

(b) Outcome under the IID setting for different amounts of data on the target client.

Figure 6: Ablation results of SFU.

**Impact of the amount of data on the target client.** The data volume of the target client reflects, to some extent, how strongly it influences the learned global model during FL training. Fig. 6(b) reports the outcome on CIFAR-10 (IID) across three architectures when varying the target client's data proportion. As the target client holds more data, its contribution is more deeply encoded in the global parameters, and the unlearning update must remove a larger portion of this contribution; this inevitably increases the tension between *forgetting* the target and *preserving* the utility for the remaining clients. Consequently, the post-unlearning test accuracy on the remaining clients decreases for all backbones as the target client becomes larger. This trend is consistent across architectures, suggesting that the degradation is primarily driven by the target client's increasing influence rather than a model-specific artifact. Nevertheless, SFU remains effective in removing the target client's contribution under almost all target-data settings, indicating that the method is robust to substantial variations in the target client's importance in the system and can operate reliably even when the target client dominates a non-trivial fraction of the training data.

## 5 Conclusion

In this paper, we propose a novel federated unlearning approach, *SFU*, that can effectively eliminate the contribution of a specified client to the global model while minimizing the utility loss on the remaining clients. The core idea is to perform the unlearning update in a constrained manner by carrying out gradient ascent within an orthogonal subspace derived from the non-target clients' gradients, which does not require restarting the entire learning process. Notably, SFU only requires access to the target client that needs to be forgotten, and does not rely on the server or other clients to store historical parameter updates or client-specific training trajectories, making it practical for real-world deployments with limited logging, realistic communication constraints, and heterogeneous client availability. We evaluate SFU under backdoor attacks to explicitly quantify whether the target client's influence has been removed, and extensive experiments across different models and settings demonstrate that SFU is both effective in forgetting and efficient in computation, achieving a favorable trade-off between unlearning strength and model utility in practice.

## Acknowledgements

Li Shen is supported by the NSFC Grant (No. 62576364), Shenzhen Basic Research Project (Natural Science Foundation) Basic Research Key Project (NO. JCYJ20241202124430041).

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

# A    Appendix

We conducted experiments on real-world datasets including MNIST, CIFAR10, and CIFAR100. The experimental settings are described in detail below.

**Dataset.**    We utilized real-world datasets, specifically MNIST, CIFAR10, and CIFAR100. The MNIST dataset [22] consists of 60,000 training samples and 10,000 test samples distributed across 10 classes. Each data sample is a grayscale image of dimensions $28 \times 28$. The CIFAR10 dataset comprises 50,000 training samples and 10,000 test samples spread across 10 classes. Each data sample is a color image with dimensions $3 \times 32 \times 32$. Similarly, the CIFAR100 dataset [21] includes 50,000 training samples and 10,000 test samples distributed among 100 classes, with 500 training samples per class. The normalization of pixel values for MNIST and CIFAR10/100 is performed using mean and standard deviation values of [0.5, 0.5, 0.5] for both.

| Datasets | Training Data | Test Data | Classes | Size |
|---|---|---|---|---|
| MNIST | 60,000 | 10,000 | 10 | $28 \times 28$ |
| CIFAR-10 | 50,000 | 10,000 | 10 | $3 \times 32 \times 32$ |
| CIFAR-100 | 50,000 | 10,000 | 100 | $3 \times 32 \times 32$ |

Table 3: Summary of dataset characteristics

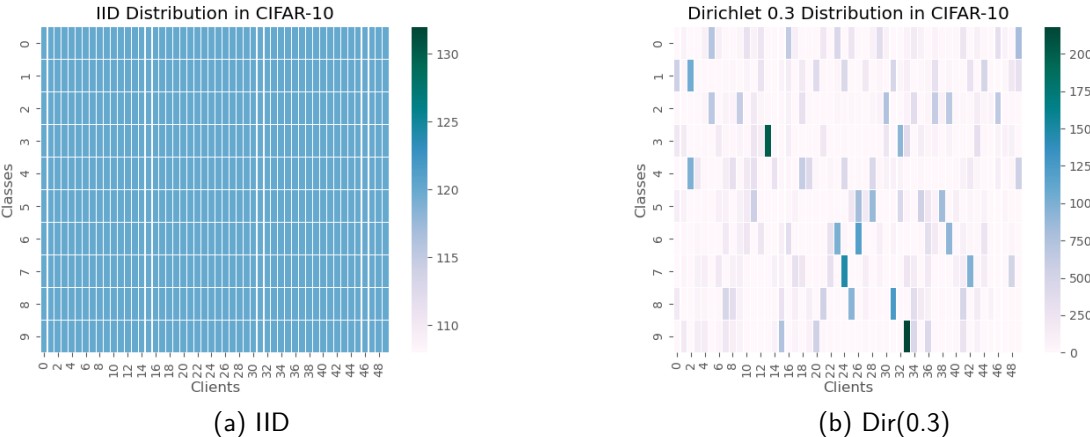

(a) IID
(b) Dir(0.3)

Figure 7: Heat maps for each client on CIFAR-10 under different data partitions. The color bar denotes the number of samples, and each cell shows the per-class sample count for a client.

**Dataset Partitions.**    To ensure a fair comparison with other baselines, we introduce heterogeneity by partitioning the total dataset based on label ratios sampled from the Dirichlet distribution. A parameter controls the level of heterogeneity in the data partition. Fig. 7 displays heat maps illustrating the label distribution across different datasets. Notably, for a heterogeneity weight of 0.3 in the Dirichlet distribution, approximately 10% to 20% of categories dominate each client, as depicted by the blue blocks in Fig. 7(b). In contrast, the IID dataset is uniformly distributed across clients, as indicated by the blue block in Fig. 7(a).

**Baseline Algorithms.**    We evaluate our proposed federated unlearning algorithm (SFU) against four baseline methods:

- **Retraining**: This method involves retraining the entire FL system without excluding the target client, resulting in computational and communication overhead.
- **UL-Subtraction** [46]: This approach forgets the target client by subtracting historical parameter updates unique to the target client from the global model. Knowledge distillation is used to alleviate the distortion caused by subtraction.
- **Unlearning via Projected Gradient Ascent (UPGA)** [15]: UPGA leverages gradient ascent information from the target client to revert the learning process and achieve unlearning, while constraining updates to an $\ell_2$-norm ball.

- **EWC-SGA** [47]: EWC-SGA employs the Fisher Information matrix to regularize the cross-entropy loss and control parameter updates, with higher importance factors imposing stricter constraints.

**Network Architectures.**    We employ three neural network architectures in our experiments:

- **MLP:** A fully-connected neural network with 2 hidden layers, containing 200 and 100 neurons, respectively.
- **CNN:** A network architecture consisting of 2 convolutional layers with 64 5×5 filters, followed by 2 fully connected layers with 800 and 500 neurons and a ReLU activation function.
- **ResNet:** We adopt a standard 4-stage ResNet with global average pooling and a linear classifier. For MNIST, we modify the first convolution to accept 1-channel inputs. We also evaluate a lightweight variant by reducing the stage widths while keeping the same topology.

All neural network architectures were implemented in PyTorch [36].

**Implementation Details.**    The hyperparameters for each method were set as follows: SFU used a learning rate of 0.01, epoch as 1, and a mini-batch size of 128 for gradient ascent on the target client. SVD parameters followed the setting of Saha et al. [38], exploring $\epsilon$ within the range $[0.90 - 0.99]$ and selecting the optimal value. UPGA and EWC-SGA maintained the same learning rate and mini-batch size as SFU. ULS on the server used a public dataset formed by randomly sampling one-tenth of the total data. For the model performance recovery experiment, FL training commenced with the stochastic model for full retraining. For SFU, UPGA, and EWC-SGA, FL training started on the unlearned local model without the target client's involvement. Knowledge distillation with the server's public data aided model accuracy recovery for UL. Additionally, parameter searches were performed to determine the best hyperparameter values.

**Unlearning Metrics**    Comparing the distinction between the unlearned model and the retrained model serves as a fundamental criterion for measuring the effectiveness of unlearning. Common dissimilarity metrics encompass model test accuracy difference [4], $\ell_2$-distance [48], and Kullback-Leibler (KL) divergence [39]. Nevertheless, in the context of Federated Learning (FL), assessing the removal of a specific client's contribution through these model difference-based evaluation methods can be challenging. Alternative metrics involve privacy leakage within the differential privacy framework [39] and membership inference attacks [13; 2]. In this study, we adopt backdoor triggers [14] as an effective means to evaluate unlearning methods, akin to Wu et al. [46]. Specifically, the target client employs a dataset containing a fraction of images with inserted backdoor triggers. Consequently, the global FL model becomes susceptible to these triggers. A successful unlearning process should result in a model that reduces accuracy for images with backdoor triggers while maintaining strong performance on regular (clean) images. It is important to note that we employ backdoor triggers solely for the purpose of evaluating unlearning methods; we do not consider any malicious clients [49; 1; 11].

**Our Evaluation Metric.**    We use backdoor attacks in the target client's updates to the global model, as described earlier, to intuitively assess the unlearning effect based on the attack success rate of the unlearned global model. In Table 1, we record the attack success rate as "atk acc". A lower "atk acc" indicates a cleaner removal of the target client's contribution. In our experiments, we implement the backdoor attack using a "pixel pattern" trigger of size 2×2 and change the label to "9". Due to the predictive errors of the model, even the attack success rate after retraining is greater than 0. We can consider the attack success rate after retraining as the benchmark for successful removal of the target client's contribution. Additionally, we use the accuracy metric on clean test data to gauge the model's performance after unlearning, denoted as "test acc" in Table 1. A high accuracy suggests that unlearning has minimal impact on the model's performance.

