# OpenReview forum: "Subspace based Federated Unlearning"
_TMLR — Accepted by TMLR_

### Review · Reviewer_3Efq · 2025-08-17

**Summary Of Contributions:**

This paper proposes a novel method for “federated unlearning,” which aims to remove the influence of a particular client’s data from a trained model in federated learning via post-processing. The proposed algorithm is based on the idea of projecting the unlearning gradient into the orthogonal subspace of the gradient directions of the remaining clients. The effectiveness of the method is also demonstrated through numerical experiments.

**Audience:**

Yes

**Audience Explanation:**

Federated learning and unlearning are both topics that have attracted considerable attention in the machine learning community in recent years, and if the claims of this paper are correct, its findings are likely to draw interest from many researchers.

**Claims And Evidence:**

No

**Claims Explanation:**

* There are numerous concerns regarding unclear points and the logical validity of the theorems, proofs, and notation.
    * In **Theorem 1**, it is unclear how $T$ is defined, and what exactly is meant by the “solution obtained through retraining.”
    * In **Assumption 1**, it is assumed that the expectation of $\|\nabla L(w)\|^2$ is bounded by $G^2$, but in the proof of Theorem 1 the inequality $\|\nabla L(w)\| \le G^2$ is used. This is inconsistent in two ways: taking expectations and taking squares.
    * In **Theorem 2**, it is unclear why $\langle g_P, g_I \rangle \le 0$ implies that $-g_P$ is a descent direction of $L(w)$. Under the natural interpretation, one would need to show that $\langle -g_P, \nabla L(w) \rangle < 0$. On the other hand, since $L(w)$ is defined using data from all clients in equation (1), its gradient does not correspond to $g_I$, so this is confusing.
    * The definition of $g$ in **Theorem 2** is unclear.
    * In **Algorithm 1** Step 14 and Section 3.2.2, $S^l = \operatorname{span}\{u_1^l,\ldots,u_k^l\}$ is defined, i.e., $S^l$ is a subspace. Then $P^l$ is defined as $P^l = S^l (S^l)^T$, but this expression does not make notational sense. Is this a typo for $P^l = U^l (U^l)^T$?
    * In **Algorithm 1** Step 15, it says $P \leftarrow [P^1, P^2, \ldots]$, but this notation looks like the matrices $P^l$ are concatenated horizontally. If the intention is to apply projection $P^l$ for each layer $l$, then $P$ should instead be a block-diagonal matrix with $P^l$ along the diagonal.
    * In **Algorithm 1** Steps 2 and 6, the notation $\nabla L_i(w^*;D^i)$ is used, but I could not find its definition in the paper. It should be clarified whether this means the gradient computed using all data in $D^i$, or a stochastic gradient obtained by sampling a subset, etc.
    * Just before equation (3), is $(R^l)_t$ a typo for $(R^l)_k$?
    * In the “Target Client” paragraph of Section 3.2.1, is the expression inside the argmin also a typo?
* Although the experimental section claims the effectiveness of the proposed method, its advantage over existing approaches is not made clear. In fact, as can be seen from Table 1, for the attack success rate existing methods such as UPGA and ULS often outperform the proposed method. From the perspective of the trade-off between test accuracy and attack success rate, it is therefore difficult to judge how advantageous the proposed method actually is.

**Requested Changes:**

I would appreciate it if you could review the concerns I have raised regarding the evidence and explain any misunderstandings on my part or possible ways to address them.

---

> ### Author Response · Authors · 2025-11-05
> **Response to Reviewer 3Efq (1/2)**
>
> Thank you very much for the careful review and the many constructive suggestions—they significantly improved the clarity and quality of our work. Below we address each point in turn; the corresponding revisions are highlighted in **blue** in the paper.
>
> ---
>
> ### **Q1. Theorem 1: definition of $T$ and “solution obtained through retraining”**
>
> At the beginning of **Sec. 3.3 (On the effectiveness of SFU)** we explicitly define:
>
> - $T$: the total number of gradient updates (i.e., the aggregated local epochs/steps) performed during the original FL training.
> - The “solution obtained through retraining,” $w_r$: the model trained from scratch without the target client’s data, using the same initialization as in the original FL run.
>
> In the same paragraph, we also clarify the remaining symbols: let $w^\*$ denote the final global model produced by the original FL training, and let $w_{ul}$ denote the model obtained by one application of SFU starting from $w^*$.
>
>
> ---
>
> ### **Q2. Assumption 1: consistency in expectations and squares**
>
> We unify the assumption and the proof by adopting a single constant $G$ that bounds expected gradient magnitudes. Concretely, we assume the expected squared norm is bounded, $\mathbb{E}\|\nabla L(w)\|_2^2 \le G^2,$  which (by Cauchy–Schwarz) implies the linear bound
> $ \mathbb{E}\|\nabla L(w)\|_2 \le G.$
>
> For consistency, we use the same linear bound with the same $G$ for the retraining objective
> $$
> L^r(w):=\sum_{i\in [N]\setminus\{I\}}\frac{|\mathcal D^i|}{|\mathcal D|}\,L_i(w),
> \qquad
> \mathbb{E}\|\nabla L^r(w)\|_2 \le G,
> $$
>
> and for the target-client gradient $\mathbb{E} \|g_I\| \le G.$ Under these bounds, the proof of Theorem 1 is rewritten in expectation form and yields
> $$
> \mathbb{E}\|w_r - w_{ul}\|_2 \le (2\eta T+\eta)\,G .
> $$
> This addresses the reviewer’s concern and removes the inconsistency in taking expectations and taking squares.
>
>
>
> ---
>
> ### **Q3. Theorem 2: why $\langle -g_P\,g_I\rangle\le 0$ gives a descent direction for $L(w)$**
> We clarify that Theorem 2 concerns the target surrogate $L_I^{\mathrm{ul}}(w):=1/L_I(w)$, not the global objective $L(w)$. The goal is to show that the update direction $-g_P$ induces forgetting by decreasing $L_I^{\mathrm{ul}}$ at $w^*$.
>
> Let $g_I:=\nabla L_I^{\mathrm{ul}}(w^\*)$,
> and let $P$ be the orthogonal projector onto the subspace spanned by the other clients’ gradients at $w^\*$. Decompose
> $$
> g_I := P g_I + (I-P)g_I,
> \qquad g_P := (I-P)g_I,
> \quad g_P \perp P g_I .
> $$
> Then
> $$
> \langle -g_P, g_I\rangle
> = \langle -g_P, g_P\rangle + \underbrace{\langle -g_P, P g_I\rangle}_{=0}
> = -\|g_P\|_2^2 \le 0,
> $$
> so $-g_P$ is a **descent direction for $L_I^{\mathrm{ul}}$** at $w^*$. Intuitively, $g_P$ is the target-unique component of the surrogate gradient; descending along $-g_P$ reduces $L_I^{\mathrm{ul}}$ (while avoiding directions shared with other clients), which captures the intended forgetting effect. We revised the wording to avoid any confusion with statements about descent for the global objective $L(w)$. *(Edits marked in blue in the manuscript.)*
>
> ---
>
> ### **Q4. Theorem 2: definition of $g$**
>
> Thank you for pointing out the ambiguity. In the original proof, the condition $\langle u, g\rangle \le 0$ used an undefined symbol $g$—it was merely a placeholder to illustrate that two opposite-direction vectors have a non-positive inner product. In the revision we use the standard descent criterion with the general gradient notation $\nabla f(w)$: a vector $u$ is a descent direction for a differentiable $f$ at $w$ if $\langle u, \nabla f(w)\rangle \le 0$. This eliminates the notational gap.
>
>
>
>
>
> ---
>
> ### **Q5. Algorithm 1 & Sec. 3.2.2: projector construction**
>
> Thank you for pointing this out. The earlier notation $P^l = S^l (S^l)^\top$ was ambiguous because $S^l$ referred to a subspace rather than an explicit basis matrix. We now construct $P^l$ via an SVD-based basis and write
> $$
> P^l := U_k^l (U_k^l)^\top,
> $$
> where $U_k^l$ contains the top-$k$ left singular vectors selected by an energy threshold. Concretely, $k$ is the smallest index such that most of the energy is retained, and $P^l$ is thus the orthogonal projector onto $\mathrm{span}(U_k^l)$. The algorithmic steps and the text in Sec. 3.2.2 have been updated accordingly (edits in **blue**).

---

> ### Author Response · Authors · 2025-11-05
> **Response to Reviewer 3Efq (2/2)**
>
> ### **Q6. Algorithm 1 Step 15: block-diagonal aggregation across layers**
>
> Thank you for the suggestion. We have corrected the aggregation to the **block-diagonal** form
> $$
> P := \mathrm{diag}(P^1,\ldots,P^L),
> $$
> so that the layerwise projectors are applied independently, as intended. We have also updated the corresponding text in **Sec. 3.2.2 (Computation of the projection matrix)** to reflect this change.
>
>
>
> ---
>
> ### **Q7. Algorithm 1 Steps 2 & 6: clarification of $\nabla L_i(w^*;\mathcal D^i)$**
>
> To avoid overloaded notation, we simplify both symbols that previously carried an explicit dataset argument. Specifically, the strings "$\nabla L_I^{ul}(w;  \mathcal D^I)$" and "$\nabla L_i(w^*; \mathcal D^i)$" did not denote different objectives; they were only meant to indicate stochastic evaluation on local data. We now drop the dataset argument and write uniformly
>
> $g_I := \nabla L_I^{ul}(w^*)$
>
> $g_i := \nabla L_i(w^*)$
>
> Both $g_I$ and $g_i$ are mini-batch (stochastic) gradients evaluated at $w^*$ on samples drawn from the respective local datasets $\mathcal D^I$ and $\mathcal D^i$.
>
> Concretely:
> - **Target client:** compute $g_I=\nabla L_I^{ul}(w^*)$ on a mini-batch from $\mathcal D^I$ and send $g_I$ to the server.
> - **Other clients:** for each $i\neq I$, compute $g_i=\nabla L_i(w^*)$ on a mini-batch from $\mathcal D^i$ and send $g_i$ to the server.
>
> We have updated **Sec. 3.2.1 (Target Client / Other Clients)** and **Algorithm 1 (Steps 2 & 6)** accordingly; edits are marked in blue.
>
>
>
>
>
>
> ---
> ### **Q8. Typo before Eq. (3): $(R^l)_t$ vs. $(R^l)_k$**
>
> Thank you for pointing this out. We have corrected the notation to $(R_k^l)$ throughout. We also state explicitly that we use the rank-$k$ approximation with the energy criterion $\|R_k^l\|_F^2 \ge \varepsilon^l \|R^l\|_F^2$, where $R_k^l$ is the best rank-$k$ approximation of $R^l$ and $\varepsilon^l \in (0,1]$ is the layer-specific energy threshold.
>
>
> ---
>
> ### **Q9. Sec. 3.2.1 Target Client: expression inside the argmin**
> Corrected: the target client **minimizes** $L_I^{\mathrm{ul}}(w)=1/L_I(w)$ to implement gradient ascent on $L_I$ using standard descent machinery (the text and Algorithm 1 have been updated accordingly).
>
> ---
>
> ### **Q10. Empirical advantage and the test–attack trade-off**
>
> To clarify SFU’s empirical advantage, we added a test–attack trade-off analysis and a scatter plot (Fig. 3, MNIST–MLP). Due to training stochasticity, even full retraining can yield a non-zero attack accuracy; therefore we take the retraining point as a reference and deem a method successful if its attack accuracy is at or below retraining (within a small tolerance). Among such points, higher test accuracy is better (upper-left region). Under this criterion, SFU lies on or near the empirical Pareto frontier: it retains the highest test accuracy while keeping attack accuracy within the acceptable range, whereas some baselines reduce attack accuracy only at the cost of substantial drops in test accuracy. We expanded the discussion around Table 1 in **Sec. 4.2 (Main Results)** to reflect this analysis.
>
>
> ---
> Once again, we appreciate the reviewer’s detailed feedback. The above revisions resolve the notational ambiguities, align the theory with the intended objectives, and clarify SFU’s empirical strengths under a principled evaluation criterion.

---

### Review · Reviewer_8CB1 · 2025-09-12

**Summary Of Contributions:**

This paper studies unlearning in a federated learning (FL) system, referred to as federated unlearning for brevity. While federated unlearning has been studied previously, existing popular methods typically require additional storage for historical data in order to remove the influence of a client that wishes to leave the FL system. This paper proposes a storage-efficient strategy for federated unlearning, called subspace-based federated unlearning.

Strengths: (i) The problem setup and method advantages are clearly described. (ii) Sufficient numerical results are provided for demonstration.

Weaknesses: The theoretical results are basic, and the technical writing could be improved.

**Additional Comments:**

N/A

**Audience:**

Yes

**Audience Explanation:**

This paper provides an interesting approach to tackling federated unlearning. The algorithm's update direction is obtained by projecting the gradient of the target client onto the orthogonal space of the gradients of the remaining clients, so that it still maximizes the targeted loss while mitigating the effects on the remaining clients.

**Claims And Evidence:**

Yes

**Claims Explanation:**

The method developed in this paper is based on projecting the gradient of the targeted client onto the orthogonal space of the gradients of the remaining clients. This approach is fundamentally different from retraining-based methods and is therefore storage-efficient.

**Requested Changes:**

Major Comments: I have some concerns regarding the theoretical results in this paper.

(i) Theorem 1 provides an upper bound for $\|w_r(T) - w_{ul}(T)\|$, where $w_r(T)$ denotes the weights obtained from training without the targeted client, and $w_{ul}(T)$ denotes the weights obtained by SFU. It uses $w(T)$, the weights obtained from training with all clients, as a reference, and establishes upper bounds for $\|w_r(T) - w(T)\|$ and $\|w(T) - w_{ul}(T)\|$, respectively.

It is unclear to me why Assumption 1 implies (7). I believe the right-hand side should be $\eta\|\nabla L_r(w_r(t)) - \nabla L(w(t))\|$. Why is this term bounded by $2\eta G$? Recall that there is no assumption made on $\nabla L_r$, and  Assumption 1 is stated in expectation.

Moreover, (9) does not imply any useful property specific to SFU. It appears that this inequality holds even without performing the projection onto the orthogonal space.

(ii) Theorem 2 is somewhat confusing. The statement claims that $-g_P$ is a descent direction for $L(w)$, i.e., the global objective over all clients. However, the last sentence of the proof states that $-g_P$ is a descent direction for client $i$'s objective. This causes confusion about which objective function the descent direction is actually associated with.

I believe this theorem can only be properly evaluated once its meaning is clarified.

Minor comments:

Please try to fix the typos in the manuscript. For example,

1. Page 6, Theorem 1. Assumptions1 should be Assumption 1

2. Pages 6-7, Proof of Theorem 1. $w_{ul}$ or $w_{ul}(T)$

3. Page 7, Display (7). What is $L^u$?

The displays on page 7 waste too much space, in my opinion. Please try to make them more compact.

---

> ### Author Response · Authors · 2025-11-05
> **Response to Reviewer 8CB1**
>
> We thank the reviewer for the careful re-evaluation and constructive feedback. Below we respond point-by-point; the corresponding revisions are highlighted in blue in the manuscript.
>
> ---
>
> ### **Q1. Why does Assumption 1 imply the bound in (7)?**
>
> We have rewritten **Assumption 1** to make all boundedness statements explicitly in expectation and to include both the retraining objective and the target-client gradient:
>
> **Assumption 1 (bounded gradients in expectation).**  For all $w$,
>
> $
> \mathbb{E}\|\nabla L(w)\|_2^2 \le G^2
> \quad\Rightarrow\quad
> \mathbb{E}\|\nabla L(w)\|_2 \le G.
> $
>
> The same bound holds for the retraining objective
> $$
> L^{r}(w) \;:=\; \sum_{i\in[N]\setminus\{I\}}\frac{|\mathcal D^i|}{|\mathcal D|}\,L_i(w),
> \quad\text{i.e.}\quad
> \mathbb{E}\|\nabla L^{r}(w)\|_2 \le G,
> $$
> and for the target-client gradient $g_I$ (defined at $w^*$ in our method), i.e. $\mathbb{E}\|g_I\|_2 \le G$.
>
> With this, using the triangle inequality,
> $$
> \begin{aligned}
> \|w_r(t{+}1)-w(t{+}1)\|_2
> &= \big\|[w_r(t)-w(t)] - \eta\nabla L^{r}(w_r(t)) + \eta\nabla L(w(t))\big\|_2 \\
> &\le \|w_r(t)-w(t)\|_2 + \eta\|\nabla L^{r}(w_r(t))\|_2 + \eta\|\nabla L(w(t))\|_2.
> \end{aligned}
> $$
> Taking expectations and applying Assumption 1 yields
> $$
> \mathbb{E}\|w_r(t{+}1)-w(t{+}1)\|_2
> \le \mathbb{E}\|w_r(t)-w(t)\|_2 + 2\eta G.
> $$
> Unrolling the recursion gives the bound used in (7):
> $$
> \mathbb{E}\|w_r(T)-w(T)\|_2 \le 2\eta T\,G .
> $$
>
> This addresses the reviewer’s concern that the right-hand side should involve gradient norms; our revision makes the expectation and constants explicit and consistent.
>
> ---
>
> ### **Q2. (9) does not imply any useful property specific to SFU**
>
> Equation (9) is not intended to, by itself, assert a property specific to SFU; it is a technical bound controlling the one-shot deviation between two models.  According to equation (4), SFU updates
> $$
> w_{\mathrm{ul}} \;=\; w(T) - \eta\, g_P,
> \qquad
> g_P := (I-P)g_I,
> $$
> where $P$ is the orthogonal projector onto the span of the remaining clients’ gradients. Hence
>
> $\|w(T)-w_{\mathrm{ul}}\|_2
> = \eta\|g_P\|_2
> \le \eta\|g_I\|_2,$
>
> and therefore
> $$
> \mathbb{E}\|w(T)-w_{\mathrm{ul}}\|_2 \le \eta\,\mathbb{E}\|g_I\|_2 \le \eta G .
> $$
> The **projection** is essential here: the inequality relies on $\|(I-P)g_I\|_2 \le \|g_I\|_2$, which is the core of SFU's update mechanism, where $g_P$ is updated through the projection. We have rewritten the proof of (9) to emphasize this fact.
>
> ---
>
> ### **Q3. Theorem 2 is confusing about which objective has the descent direction**
>
> Thank you for pointing this out. We have clarified the statement and proof. The primary goal of Theorem 2 is to demonstrate that the update direction $-g_P$ will induce **forgetting** for the target client by reducing the target client's surrogate loss $L_I^{ul}$.
>
> In particular, we now show that $g_I$, the gradient of the surrogate $L_I^{ul}$, would make $L_I^{ul}$ increase. However, $-g_P$ provides the descent direction that drives $L_I^{ul}$ downward. This is the core of the “forgetting” mechanism.
>
> Theorem 2 establishes this by proving that $-g_P$ is a descent direction for $L_I^{ul}$ at $w^*$, and this is because
> $$
> \langle -g_P\, g_I \rangle = -\|g_P\|_2^2 \le 0.
> $$
> This inequality shows that $-g_P$ reduces the surrogate loss $L_I^{ul}$ at the target client and thus achieves the forgetting effect.
>
> We have rewritten the final statement of the proof to make this clearer and avoid any confusion with the global objective $L(w)$, which was misleading in the original wording.
>
>
> ---
>
> ### **Q4. Minor comments**
>
> - We fixed typos, e.g., “Assumptions1” → “Assumption 1”, and unified notation for $w_{ul}$.
> - The  symbol  $L^{ul}$ in “Display (7)” has been removed; only $L$ and $L^{r}$ are used.
> - Long displays on page 7 were compacted; ancillary explanations are moved into the running text for better readability.
>
> ---
>
> Once again, we appreciate the reviewer’s insightful suggestions. They helped us significantly improve the rigor and clarity of our theory section and its presentation.

---

### Review · Reviewer_B2Ph · 2025-10-23

**Summary Of Contributions:**

The paper introduces Subspace-based Federated Unlearning (SFU), which is a new method to remove a client’s influence from a federated learning (FL) model without retraining or storing historical updates. SFU performs gradient ascent on the target client but constrains it to an orthogonal subspace to the gradient space of other clients. This helps erase the target client’s contribution while preserving overall model performance. The method requires only one local training round per client and no storage of historical gradients, making it practical for resource-constrained FL systems. The paper provides theoretical guarantees and empirical experiments on MNIST, CIFAR-10, and CIFAR-100. SFU achieves strong unlearning performance (reducing attack success rates) while retaining model accuracy better than prior baselines (UPGA, EWC-SAG, ULS).

Key Strengths of the paper
1. The paper is well written to explain the idea clearly and thoroughly.
2. The proposed SFU is very efficient given that it requires only one local training round per client and no storage of historical gradients.
3. The paper provides a good theoretical foundation with convergence and effectiveness proofs.

Weaknesses of the papaer
1. The empirical experiments need improvement. The accuracy of CIFAR10 models are typically below 50% which is unacceptable. Even under the Federated Learning scenario, with 10 clients, I think it's reasonable to expect the accuracy to be ~80%. The extremely low accuracy may suggest either that the model is too simple or that the training process is not correct. This will weaken the readers' belief of the effectiveness of the experiments.
2. The paper should state more clearly regarding the resource usage of SFU vs. the other method. In my understanding, the SFU will require the unlearning client to possess the training data (or at least the data that the client wants to forget). While other unlearning methods may not require this data but needs to store the historical gradient data on server end. I don't think this is a disadvantage itself; instead, this could be an advantage that clients can determine what data they want to unlearn from the model. However, I think the author should state more clearly regarding the resource requirements of their methods.

**Audience:**

Yes

**Audience Explanation:**

The idea presented in this paper is novel and well written. Subspace-based Federated Unlearning (SFU) offers a new and efficient approach to address an important problem in federated unlearning. Given the growing interest in privacy, unlearning, and resource-efficient federated learning, this method would be of broad interest to the TMLR audience, particularly those working on model unlearning, privacy-preserving machine learning, and distributed optimization. The clear presentation and strong theoretical grounding make the findings valuable for researchers pursuing similar directions.

**Broader Impact Concerns:**

No, I don't find any concerns about the ethical implications of the work.

**Claims And Evidence:**

No

**Claims Explanation:**

The major issue is with the results of empirical experiments. I think the extremely low test accuracy of the trained CIFAR10 (<50%) and CIFAR100 (~20%) models will make the audience doubt the reliability of the results. Because this may suggest either that the model is too simple or that the training process is not correct.

**Requested Changes:**

1. Improve experimental results (Critical). The current CIFAR-10 models achieve unusually low accuracy (often below 50%), which raises concerns about the validity of the training setup or model capacity. The authors should revisit the experimental design to ensure models are properly selected and trained, and results are representative of realistic FL performance (expected 80% accuracy).
2. Clarify resource requirements (Important but not critical). The paper should more clearly discuss the resource implications of SFU compared to other methods. Specifically, SFU requires the target client to have access to its data during unlearning, while other methods store historical gradients on the server. This clarification would help readers fairly assess the trade-offs in practicality and deployment.

---

> ### Author Response · Authors · 2025-11-05
> **Response to Reviewer B2Ph**
>
> We sincerely thank you for the careful review and constructive suggestions. We address the two points below.
>
> ### **Q1. Improve experimental results**
>
> As noted in the Appendix (Network Architectures), our initial submission used a 3-layer MLP and a 5-layer CNN under resource constraints. These backbones, even in centralized settings, typically do not reach 80% accuracy on CIFAR-10. However, all methods were evaluated under the same model and budget, ensuring fair comparisons between them.
>
> We also acknowledge your concern about the accuracy and have added a stronger backbone (ResNet) to better align with typical federated learning benchmarks. Under the federated setting with 10 clients, the IID CIFAR-10 accuracy now meets the expected range (~80%). The updated results are as follows:
>
> | Distribution | FedAvg (test/atk) | Retraining (test/atk) | UPGA (test/atk) | ULS (test/atk) | EWC-SGA (test/atk) | **SFU (test/atk)** |
> |--------------|-------------------|-----------------------|-----------------|----------------|-------------------|--------------------|
> | **IID**      | 83.73 / 38.27      | 83.98 / 7.23          | 76.84 / 8.56    | 12.61 / 0.00   | 77.53 / 0.84      | **78.06 / 0.46**   |
> | **Dir(0.6)** | 80.79 / 46.66      | 81.62 / 2.49          | 75.40 / 7.52    | 67.44 / 0.00   | 71.64 / 0.94      | **77.95 / 2.82**   |
> | **Dir(0.3)** | 79.91 / 56.87      | 79.55 / 1.59          | 50.35 / 0.08    | 13.07 / 0.00   | 65.04 / 2.31      | **71.01 / 1.56**   |
>
>
> These results show that, with the ResNet backbone, CIFAR-10 accuracy is now within a realistic range. SFU achieves a strong test–attack accuracy trade-off, often performing at or below the retraining attack accuracy, while still maintaining competitive clean accuracy. We have included the complete experimental results in **Table 1**, and detailed architecture settings are provided in the **Appendix** (highlighted in blue).
>
>
> ---
>
> ### **Q2. Clarify resource requirements**
>
> Thank you for your thoughtful suggestion. We agree that it is important to clarify the resource implications of SFU in comparison to other methods. To that end, we have added a **comparison of resource, communication, and time costs** across different unlearning methods, now detailed in **Sec. 4.3 (Resource Implications and Deployment Trade-offs)**.
>
>
> We also acknowledge your point that SFU requires the unlearning client to possess the training data (or at least the data it wishes to forget). We do not view this as a disadvantage. In fact, this design provides the client control over what data to unlearn, enabling fine-grained, client-controlled forgetting (e.g., selectively unlearning specific subsets of data). Moreover, this approach eliminates the need for server-side storage of historical gradients, which is particularly beneficial in scenarios where long-term storage of client histories is prohibited or impractical. Additionally, SFU’s computational cost can be reduced by sampling a subset of non-target clients, which further enhances efficiency in specific deployment scenarios.
>
>
>
> We have added this discussion in **Sec. 4.3**, together with a comparison table summarizing the resource trade-offs across methods; new content is highlighted in blue.
>
>
>
> ---
>
>
>
> We appreciate your valuable feedback. With the **ResNet-based experiments** and the **resource/trade-off discussion**, we believe we have addressed the two main concerns while maintaining the fairness of the comparison and supporting our claims with clear, revised results.

---

### Author Response · Authors · 2025-11-14
**Author Response Update**

We would like to once again thank all reviewers for their thoughtful and constructive feedback, and for the time and effort invested in helping us improve this work.

In response to your valuable suggestions, we have made the following updates:
- Conducted additional **ResNet-based experiments** and **further Test–Attack accuracy trade-off analysis** in Section 4.2;
- Expanded the discussion on **resource usage and deployment trade-offs** in Section 4.3;
- Further **refined and clarified the theoretical analysis** in Section 3.

All major revisions are highlighted in **blue** in the updated manuscript.

If any reviewer has further questions, remaining concerns, or points requiring clarification, we would greatly appreciate your continued feedback. We sincerely look forward to hearing your thoughts.

---

### Decision · Action_Editor_EyRB · 2025-11-30

**Recommendation:** Accept with minor revision

**Additional Comments:**

Please update Figures 5 and 6 to incorporate the ResNet results.

**Audience:**

Yes

**Audience Explanation:**

TMLR's audience, particularly researchers in Federated Learning and machine unlearning, would be highly interested in the findings, as the paper proposes a novel, storage-free, and communication-efficient method (SFU) for federated unlearning that effectively balances unlearning success and model utility preservation.

**Claims And Evidence:**

Yes

**Claims Explanation:**

The claims made in the submission are indeed supported by accurate, convincing, and clear evidence, which is presented through a combination of rigorous theoretical analysis, extensive empirical evaluation, and insightful ablation studies.

---

> ### Author Response · Authors · 2025-12-31
>
> Dear Action Editor EyRB,
>
> Thank you for your efforts in helping improve the quality of our manuscript. We sincerely appreciate your positive evaluation of our submission and the recommendation of acceptance with minor revision.
>
> Following your suggestion, we have prepared the camera-ready version and updated Figures 5 and 6 to incorporate the ResNet results. All requested changes have been incorporated into the revised manuscript accordingly.
>
> Thank you again for your time and consideration.
>
> Sincerely,
>
> The Authors